# Beyond Hotspot Mutations: Diagnostic Relevance of High Frequency, Low Frequency, and Disputed *rpoB* Variants in Rifampicin-Resistant *Mycobacterium tuberculosis*

**DOI:** 10.3390/pathogens15010016

**Published:** 2025-12-22

**Authors:** Siti Soidah, Toto Subroto, Irvan Faizal, Muhammad Yusuf

**Affiliations:** 1Doctoral Program in Biotechnology, Graduate School, Universitas Padjadjaran, Bandung 40132, Indonesia; siti19070@mail.unpad.ac.id; 2Research Center for Molecular Biotechnology and Bioinformatics, Universitas Padjadjaran, Bandung 40132, Indonesia; m.yusuf@unpad.ac.id; 3Department of Chemistry, Faculty of Mathematics and Natural Sciences, Universitas Padjadjaran, Sumedang 45363, Indonesia; 4Research Cente for Vaccine and Drugs, National Research and Innovation Agency Republic of Indonesia, Tangerang Selatan 15314, Indonesia; irvan.faizal@atmajaya.ac.id; 5School of Bioscience, Technology, and Innovation, Atma Jaya Catholic University of Indonesia, Tangerang 15345, Indonesia

**Keywords:** *Mycobacterium tuberculosis*, *rpoB* mutations, rifampicin resistance, disputed mutations, molecular diagnostics, whole-genome sequencing

## Abstract

Rifampicin-resistant tuberculosis (RR-TB) remains a major threat to global TB control, primarily driven by mutations in the *rpoB* gene of *Mycobacterium tuberculosis* (Mtb). Most resistance-conferring mutations occur within the 81-base pair RIF resistance determining region (RRDR), particularly at codons S450L, H445Y/D, and D435V, which are strongly associated with high level resistance. However, increasing evidence of low-frequency and disputed variants both within and beyond the RRDR reveals a broader genetic spectrum that contributes to diagnostic uncertainty and variable phenotypic outcomes. This review summarizes current knowledge of high frequency, low frequency, and disputed *rpoB* mutations and their implications for molecular detection of RIF resistance. Structural analyses show that specific amino acid substitutions alter key hydrogen bonds or create steric hindrance in the RIF-binding pocket, leading to diverse resistance levels. Despite the success of molecular platforms such as Xpert MTB/RIF and line probe assays, their hotspot-based detection limits sensitivity to noncanonical variants. Lowering the minimum inhibitory concentration (MIC) breakpoint and integrating sequencing-based approaches, such as targeted and whole-genome sequencing, can enhance detection accuracy. A combined genomic and phenotypic framework will be essential to close existing diagnostic gaps and advance precision guided management of RIF-resistant and multidrug-resistant tuberculosis.

## 1. Introduction

Tuberculosis (TB), caused by *Mycobacterium tuberculosis* (Mtb), remains a major global health challenge, particularly due to the rise in antimicrobial resistance. Among the drug-resistant forms of TB, rifampicin-resistant TB (RR-TB) poses the most critical threat, as rifampicin (RIF) is one of the first-line anti-TB drugs [1,2,3]. Resistance to this drug is closely linked to multidrug-resistant TB (MDR-TB) [3,4,5,6] making its rapid and accurate detection essential for effective treatment and control. The World Health Organization (WHO) estimates that approximately 400,000 cases of RR/MDR-TB occur annually, underscoring the urgent need for timely and precise diagnostic methods to guide appropriate treatment strategies [2]. Addressing this challenge requires a comprehensive understanding of the genetic basis of RIF resistance, which is central to the development of reliable molecular diagnostics and effective therapeutic interventions.

RIF resistance in Mtb is primarily driven by mutations in the *rpoB* gene, which encodes the β-subunit of RNA polymerase which is the molecular target of RIF. These mutations alter the conformation of the RIF-binding pocket, thereby reducing drug affinity and disrupting inhibition of RNA synthesis [7,8,9,10]. More than 95% of RIF-resistant strains harbour mutations within an 81-base pair region known as the rifampicin resistance-determining region (RRDR), which spans codons 426–452 in the Mtb H37Rv reference genome (equivalent to codons 507–533 according to the *Escherichia coli* numbering system for *rpoB*) [11,12,13]. This region has long been considered the central hotspot of resistance, serving as the basis for most molecular diagnostic platforms (Figure 1). However, the distribution and prevalence of specific mutations within the RRDR vary markedly across geographic regions [14,15,16,17,18,19,20,21,22,23,24,25,26,27,28,29] and are often linked to dominant circulating bacterial lineages and local treatment histories [30]. While high-frequency mutations such as S450L and H445Y are strongly associated with high-level resistance [31,32], other substitutions occur less commonly or exhibit borderline effects on drug susceptibility, making them more difficult to detect using conventional phenotypic or molecular methods [33,34].

Despite significant advances in molecular diagnostics such as Xpert MTB/RIF and line probe assays (LPA) which enable rapid detection of resistance-conferring mutations directly from clinical samples, these assays primarily target well-characterized mutations within the RRDR [11,35,36,37,38]. Their diagnostic sensitivity can be affected by the genetic diversity of circulating isolates, particularly in settings where rare or disputed mutations are more prevalent. In some cases, such mutations may yield false-susceptible results in automated phenotypic systems such as the Mycobacteria Growth Indicator Tube (MGIT), leading to underdiagnosis and suboptimal treatment outcomes [33,34,39,40]. The inability to detect RIF resistance accurately can result in inappropriate treatment regimens, misclassification of patients as drug-susceptible, delayed initiation of therapy, prolonged infectiousness, and increased community transmission. Inadequate resistance detection also contributes to poor clinical outcomes and facilitates the spread of drug-resistant strains, ultimately undermining TB control efforts at both individual and population levels [41,42].

Beyond the well-characterized mutations within the RRDR, recent evidence has revealed additional mutations outside this region [14,21,23,24,25,27,43,44,45] as well as several disputed mutations within the RRDR [14,15,16,17,19,20,22,23,25,27,29,44,46,47]. These disputed variants are often undetected by conventional DST methods, particularly automated systems such as MGIT 960, due to their variable impact on rifampicin minimum inhibitory concentrations (MICs) [33,39]. Such mutations can alter RIF binding and contribute to varying levels of resistance. Although these variants occur less frequently, they hold significant diagnostic importance because conventional probe-based assays often fail to detect them.

Recent advances in sequencing technologies, including whole-genome sequencing (WGS) and targeted next-generation sequencing (tNGS), have greatly enhanced the capacity to detect, and characterize *rpoB* variants [48,49,50,51]. These technologies have revealed the full spectrum of resistance-associated mutations and their structural consequences on the RNA polymerase β-subunit. However, their routine application in clinical diagnostics remains limited due to high costs, technical complexity, and the need for specialized bioinformatics infrastructure. As a result, their use is often confined to reference laboratories and research settings rather than point-of-care applications.

This review aims to summarize the current understanding of high frequency, low frequency, and disputed *rpoB* mutations associated with RIF resistance, to highlight their diagnostic implications, and to discuss how expanding beyond hotspot-focused detection can improve the accuracy of molecular diagnostics and the management of RR/MDR-TB.

## 2. Data Sources and Compilation Approach

This narrative review was primarily informed by the WHO Mtb mutation catalogue and key peer-reviewed studies reporting *rpoB* mutation frequencies and their clinical relevance. Additional literature was incorporated to support the structural, diagnostic, and phenotypic interpretations summarized in this work.

## 3. The *rpoB* Gene and Rifampicin Resistance Mechanism

RIF resistance in Mtb primarily results from mutations in the *rpoB* gene that alter the structure of the RNA polymerase β-subunit, a key component of the transcription elongation complex and the primary target of RIF binding. In addition to this target-site alteration, secondary mechanisms such as reduced cell wall permeability and increased efflux pump activity further decrease intracellular RIF concentration, enhancing bacterial survival and contributing to varying resistance levels [7,52]. The principal molecular determinant of resistance is mutation in the *rpoB* gene, which encodes the β-subunit of RNA polymerase [9,53,54,55]. In RIF-susceptible strains, RIF binds to this subunit and blocks the elongation of RNA chains, resulting in bacterial growth inhibition. However, mutations in *rpoB* alter the conformation of the RIF-binding pocket, disrupting the stable interaction between the drug and the enzyme and allowing transcription to continue despite the presence of RIF [7,8,54] (Figure 2). These mutations, which are concentrated within the 81 bp RRDR, account for more than 95% of RIF-resistant isolates and include common substitutions such as S450L, H445Y, and D435V [37,56,57].

Beyond conformational changes in the RIF-binding pocket, cell-wall permeability also plays a key role in limiting antibiotic entry and reducing drug susceptibility. Several proteins and genes contribute to this mechanism. PE11 (LipX/*Rv1169c)*, a member of the PE protein family that has esterase activity, alters cell wall lipid composition and increases surface hydrophobicity, thereby restricting RIF penetration [58,59,60]. Similarly, the *virS* (*Rv3082c*) regulated *mymA* operon (*Rv3083*) maintains cell wall integrity and mycolic acid composition under stress. Mutations in these genes disrupt mycolic acids and increase cell wall permeability, allowing antibiotic penetration [61,62,63]. CpnT (*Rv3903c*) is a channel-forming protein involved in nutrient uptake and toxin export also affects permeability, as cpnT mutants display elevated minimum inhibitory concentration (MICs) to multiple antibiotics [64,65].

Another important mechanism contributing to RIF resistance in Mtb is the active efflux of the drug through membrane transporters. Efflux pumps reduce the intracellular concentration of RIF, thereby enhancing bacterial survival and facilitating the emergence of resistance [7,66]. As reported by Pang et al. [52] overexpression of *Rv0783* and *Rv2936* results in elevated RIF MICs, supporting their roles as RIF-associated efflux pumps that contribute to low-level resistance. Additional efflux transporters such as *Rv2333*, *drrB*, *drrC*, *Rv0842*, *bacA*, and *efpA* may also participate in this process, acting synergistically with *rpoB* mutations. Notably, the use of efflux inhibitors such as verapamil (VP) and chlorpromazine (CPZ) has been shown to significantly restore RIF susceptibility in RIF-monoresistant Mtb strains, underscoring the combined contribution of target-site alteration and active efflux mechanisms in RIF resistance [67].

Given that *rpoB* mutations remain the primary determinant of RIF resistance, understanding their specific locations, frequencies, and phenotypic effects is critical for accurate molecular detection and clinical interpretation.

## 4. Classification of *rpoB* Mutations

The WHO-endorsed Mtb mutation catalogue [68] established a global framework for interpreting resistance-associated mutations through the analysis of over 38,000 isolates from 45 countries. Each mutation was evaluated using statistical parameters, including odds ratio, positive predictive value, FDR-corrected *p* value, and confidence interval, and it subsequently assigned to one of five confidence categories: (1) associated with resistance, (2) associated with resistance–interim, (3) uncertain significance, (4) not associated with resistance–interim, and (5) not associated with resistance (consistent with susceptibility). This confidence grading system provides a standardized reference for interpreting molecular drug susceptibility results and refining diagnostic algorithms for *rpoB*-mediated RIF-resistance (Figure 3). However, despite its broad global dataset, the WHO-endorsed Mtb mutation catalogue has several limitations, including uneven geographic representation and the inability to capture newly emerging or region-specific mutations.

Based on this WHO classification framework, several *rpoB* mutations, particularly S450L, H445Y, and D435V, consistently show high confidence associations with RIF resistance [69] and form the core group of high frequency mutations discussed in the following section. In contrast, several low frequency resistance-associated variants, and several disputed mutations also contribute to RIF resistance. Low frequency mutations can occur within or outside the RRDR and show limited global prevalence, whereas disputed mutations exhibit variable phenotypic effects and may lead to inconsistent results in routine diagnostic testing. These categories and their corresponding codon positions are visually summarized in Figure 1, which maps high frequency, low frequency, and disputed mutations across the *rpoB* gene, including sites both within and outside the RRDR.

### 4.1. High Frequency Mutations

High frequency mutations within *rpoB* represent the most consequential determinants of RIF resistance in Mtb. These recurrent variants consistently emerge across diverse geographic settings and exert strong functional effects on the RIF–RNA polymerase interaction, leading to substantial increases in drug MICs and reliable phenotypic resistance [56,70]. Because of their widespread occurrence and robust association with high-level resistance, these mutations form the central focus of diagnostic assays and remain the most informative markers for detecting RR-TB.

Findings from 29 studies consistently show that RIF resistance is predominantly explained by a small number of high-frequency *rpoB* mutations, most notably S450L, H445Y/D, and D435V, which together represent the vast majority of resistant isolates worldwide [71]. Other substitutions at these codons, such as H445L and D435Y are generally classified as disputed mutations [33,40]. Among these, S450L stands out as the most frequently detected and epidemiologically widespread variant, making it a key molecular marker for RIF resistance surveillance and diagnostic assay design [14,15,16,17,18,19,21,22,23,24,25,26,27,28,29,44,46,47].

Figure 4 highlights the dominance of the S450L mutation among RIF-resistant isolates, with H445Y, H445D, and D435V contributing smaller but notable proportions, while low frequency and disputed mutations collectively account for a minority of cases.

Structural modelling analysis have provided mechanistic insights into why the *rpoB* mutations S450L, H445Y/D, and D435V are the most prevalent and diagnostically relevant variants among RIF-resistant Mtb isolates. These mutations occur at residues that form critical hydrogen-bonding and steric interaction networks within the RIF-binding pocket of the RNA polymerase β-subunit [31,52].

Substitution at S450 disrupts a key intermolecular hydrogen bond between serine and RIF and introduces a hydrophobic leucine residue that causes steric hindrance, thereby markedly reducing drug affinity while preserving the overall structural integrity and catalytic function of the polymerase [31,52]. Similarly, mutations at H445 disrupts an intermolecular hydrogen bond between histidine and rifampicin and introduces an aromatic ring that sterically interferes with rifampicin binding, while H445D replaces a neutral histidine with a negatively charged residue, weakening hydrogen bonding and destabilizing drug interactions. At D435V, the substitution of the negatively charged aspartate with a bulkier nonpolar valine residue causes steric hindrance within the rifampicin-binding pocket, restricting access of the drug to its target site and thereby reducing binding affinity [31].

Consistent with these structural observations, phenotypic analyses have shown that the degree of RIF resistance conferred by *rpoB* mutations depends on the specific amino acid substitution. In multivariate models, S450L, H445Y, and H445D are strongly associated with high-level rifampicin resistance, whereas D435V is typically linked to moderate-level resistance, consistent with its more localized steric effect that partially restricts drug access without completely abolishing binding [31]. According to the WHO-endorsed Mtb mutation catalogue, the *rpoB* mutations S450L, H445Y, H445D, and D435V are classified as “associated with resistance”, supported by strong statistical associations and consistent phenotypic evidence across global datasets, confirming their role as key molecular markers of RIF-resistant Mtb [68].

### 4.2. Low-Frequency Mutations

Although the majority of RIF-resistant Mtb isolates harbour mutations within the RRDR, a smaller subset exhibits rare substitutions that occur at low frequencies but are consistently associated with resistance [32]. Despite their infrequency, these mutations contribute to the genetic diversity of *rpoB* mediated RIF resistance and have been reported across diverse epidemiological settings [13,28,32,57,72]. These mutations can arise both within and outside the RRDR, differing primarily in their location within the *rpoB* gene, the extent to which they affect RIF-binding affinity, and their relative prevalence among global resistant isolates. The occurrence of these mutations is also region-dependent, reflecting differences in local Mtb lineages and diagnostic coverage [73,74,75]. The distribution of selected low frequency and disputed *rpoB* mutations associated with RIF-resistance is summarized in Figure 5.

Low frequency mutations, although rare, are important to recognize because probe-based assays such as Xpert MTB/RIF, Xpert MTB/RIF Ultra, and LPA primarily target common RRDR mutations [35,76,77] and may therefore miss these less frequent variants, potentially resulting in incorrect classification of RIF resistance as susceptibility. Similarly, recent evaluations of the Xpert MTB/RIF Ultra demonstrated that uncommon or borderline *rpoB* mutations, including those located at the RRDR margins, can yield indeterminate or false susceptible results, underscoring limitations of current diagnostic platforms [78]. Moreover, certain phenotypically resistant yet genotypically occult variants have been described, remaining undetected by rapid molecular tests due to their absence within predefined probe targets [79]. These findings highlight the importance of expanded genomic surveillance and the development of adaptive diagnostic algorithms capable of capturing the full spectrum of *rpoB* mutations associated with rifampicin resistance.

### 4.3. Disputed Mutation

Disputed mutations represent a diagnostically challenging subset of *rpoB* variants that are genotypically associated with RIF resistance yet often yield phenotypically susceptible or borderline results in conventional drug susceptibility testing (DST). These mutations typically produce low-to-intermediate increases in RIF minimum inhibitory concentrations (MICs), sometimes below the critical concentration used in automated phenotypic assays such as MGIT 960, leading to false susceptible interpretations. These mutations are often missed by DST, particularly when using the automated MGIT 960 system, due to their variable impact on RIF MICs and the proximity of their MIC values to the established critical concentration [33,39,80]. As a result, patients harbouring Mtb strains with disputed mutations may be misclassified as RIF-susceptible, which poses a risk for inadequate treatment and continued transmission of resistant bacilli [40].

Beyond phenotypic misclassification, disputed *rpoB* mutations also present diagnostic challenges at the molecular level. Variants such as L430P, D435Y, H445N, H445L, L452P, I491F [33,39,40,80] confer low-level or borderline resistance, which may lead to inconsistent hybridization signals in molecular assays. Some disputed mutations may even escape detection by platforms such as Xpert MTB/RIF and LPA, particularly when located outside the RRDR (e.g., I491F) [81]. Failure to accurately identify these mutations has contributed to ongoing transmission in regions such as Suriname, Eswatini, KwaZulu-Natal, and Botswana, where borderline-resistant strains have become key drivers of RIF-resistant TB [82,83,84].

Nevertheless, recent studies have demonstrated that Xpert MTB/RIF can detect certain borderline mutations, including H445N, L430P and L452P, with higher sensitivity than phenotypic methods such as Löwenstein–Jensen (L-J) culture or MIC testing at current breakpoints [85]. Interestingly, the same study also reported the I491F mutation, which produced a RIF-resistant result on L-J culture but was not detected by Xpert MTB/RIF, confirming that mutations outside the RRDR remain a significant diagnostic blind spot [85].

Several studies have shown that isolates harbouring these disputed mutations often display rifampicin MIC values ≤ 1.0 µg/mL, near or below the WHO-defined critical concentration, resulting in discordance between genotypic and phenotypic testing [40,80,86]. Despite these borderline MICs, clinical evidence indicates that patients infected with such strains may still experience treatment failure or relapse when managed with standard rifampicin-containing regimens [42]. To ensure accurate detection and guide appropriate therapy, confirmatory sequencing or MIC-based testing is therefore strongly recommended, particularly in regions where these mutations are prevalent. Moreover, extending the incubation period of L-J medium-based DST to six weeks or lowering the MIC critical concentration to 0.5 µg/mL has been shown to improve detection of borderline rifampicin resistance [85]. Collectively, these findings underscore the need to strengthen molecular diagnostic capacity through the broader implementation of rapid assays such as Xpert MTB/RIF to improve the detection accuracy and accessibility of RIF resistance globally. An integrated overview of *rpoB* mutations, including WHO confidence categories, global frequencies, and associated levels of RIF-resistance, is summarized in Table 1.

## 5. Structural and Diagnostic Implications of *rpoB* Mutations

### 5.1. Structural Impact on RIF Binding

Structural molecular modelling studies have shown that RIF interacts with key residues Ser450, His445, and Asp435 via hydrogen bonds and hydrophobic contacts that stabilize the drug enzyme complex. Substitutions at these sites, such as S450L, H445Y/D/L, and D435V/Y, either remove crucial hydrogen bonds or introduce steric hindrance, reducing RIF affinity while preserving enzyme function [25,31], as illustrated by the structural models in Figure 6.

Mutations outside the canonical 81 bp RRDR, including L430P, I491F, and V170F, cause more subtle conformational shifts in adjacent α-helices and β-sheets, leading to reduced binding efficiency and low-level resistance. Together, these structural differences explain the spectrum of resistance phenotypes observed clinically from high level resistance with canonical mutations (e.g., S450L) to borderline or low level resistance caused by disputed variants (e.g., L430P, I491F) [31].

Although structural modelling has provided valuable insights into how key *rpoB* mutations disrupt RIF binding, current approaches remain limited by several methodological constraints. Most studies rely on static crystal structures of RNA polymerase, which do not capture the dynamic conformational changes induced by individual mutations. As a result, structural interpretations often focus only on local alterations, such as changes in hydrogen bonding, hydrophobic contacts, or steric hindrance, while overlooking broader effects on the flexibility and geometry of the RNA polymerase active pocket. Even studies that integrate MIC data with structural predictions typically assess static interactions and therefore cannot fully explain the spectrum of borderline or discordant resistance phenotypes.

Another important limitation is that existing structural models rarely incorporate the contribution of compensatory mutations in *rpoA* or *rpoC*, nor do they integrate clinical outcome data, despite evidence that these factors significantly modulate bacterial fitness and treatment response. These observations are consistent with recent experimental findings demonstrating that several *rpoB* non-RRDR mutations exert compensatory effects on RNA polymerase function and bacterial fitness, particularly by restoring transcriptional efficiency in strains carrying high fitness cost RRDR mutations such as S450L [72]. These gaps reduce the ability of structural modelling to predict resistance levels for disputed or low frequency mutations.

Future research should therefore adopt molecular dynamics–based approaches, make use of higher resolution RNA polymerase structures, and integrate structural modelling with phenotypic MIC distributions and clinical outcome datasets. Such advances will be critical for improving mechanistic understanding of *rpoB*-mediated resistance—particularly for disputed mutations and may ultimately inform the refinement of molecular diagnostics and the development of more precise therapeutic strategies.

### 5.2. Diagnostic Methodologies and Analytical Performance

A variety of diagnostic platforms are used to detect rpoB mutations associated with RIF resistance, each with different analytical targets, turnaround times, sensitivities, and detection limits. Probe-based assays such as Xpert MTB/RIF, Xpert Ultra, and LPA primarily interrogate the 81 bp RRDR and therefore show excellent performance for common high-frequency mutations but reduced sensitivity for low-frequency, disputed, or non-RRDR variants. Phenotypic DST methods, including MGIT 960 and Löwenstein–Jensen proportion tests, are affected by MIC variability and may underestimate borderline resistance. Sequencing-based tools such as tNGS and WGS overcome these limitations by providing complete coverage of *rpoB*, enabling detection of both canonical and rare variants. A comparative overview of these methodologies and their analytical characteristics is presented in Table 2.

Each diagnostic platform has distinct advantages and limitations. Xpert assays provide rapid RRDR detection but may miss non-RRDR or borderline mutations. LPAs offer mutation-specific resolution yet require higher bacterial loads, while phenotypic DST may underestimate borderline resistance due to MIC variability. Sequencing-based methods (tNGS/WGS) offer the most comprehensive coverage and are recommended when results are indeterminate, discordant, or when non-RRDR mutations are suspected.

### 5.3. Diagnostic Implications and Limitations

Molecular assays such as Xpert MTB/RIF, Xpert Ultra, and LPA detect RIF resistance by targeting mutations within the RRDR of the *rpoB* gene [35,76,77]. Although these platforms show high accuracy for canonical RRDR mutations, Xpert-based assays may still yield false-positive RIF resistance results when probe hybridization is affected by silent *rpoB* mutations, atypical amplification patterns caused by very high or very low MTB burden, or cross reactivity with non-tuberculous mycobacteria [96]. Conversely, False negative resistance results can also occur if RIF resistance mutations are present outside the RRDR, as documented for mutations such as Leu533Pro [97] and Val170Phe [96]. Taken together, these molecular limitations indicate that ambiguous or discordant Xpert results may still reflect underlying RIF resistance and warrant confirmatory testing with phenotypic DST or sequencing [78].

In addition to these molecular challenges, several borderline or disputed *rpoB* mutations pose further diagnostic difficulties because their phenotypic expression of resistance is subtle and may fall near current interpretive thresholds. Similarly, phenotypic assays such as MGIT 960 may underestimate RIF resistance when MIC elevations are modest or when mycobacterial growth is slow.

In addition to these molecular challenges, borderline and disputed *rpoB* mutations frequently generate inconsistent diagnostic outcomes. Recent clinical evidence illustrates this variability: Haas et al. reported several cases in which Xpert, LPA, WGS, and phenotypic DST produced discordant classifications of RIF resistance—ranging from Xpert-positive results without detectable RRDR mutations to low-abundance S450L or non-RRDR mutations missed by phenotypic DST. These observations highlight that ambiguous or conflicting assay results may still indicate underlying RIF resistance and should prompt confirmatory sequencing or extended phenotypic testing [96].

To improve the detection of borderline RIF resistance, recent studies have suggested extending the incubation period of L-J DST based to six weeks or lowering the MIC critical concentration to 0.5 µg/mL, which increases the likelihood of identifying low-level resistant isolates [85]. These diagnostic refinements, combined with the integration of sequencing-based approaches such as tNGS or WGS, can ensure comprehensive.

Future diagnostic strategies should expand beyond hotspot-based assays to include region-specific and lineage-associated *rpoB* variants. Integrating structural, genotypic, and phenotypic data into diagnostic algorithms will improve resistance interpretation and guide personalized therapy. The WHO mutation catalogue [68] provides a foundation for this integration, but continuous updates and genomic surveillance remain essential as new variants and mechanisms of resistance emerge.

## 6. Clinical and Epidemiological Relevance of *rpoB* Mutations

*RpoB* mutations hold substantial clinical and epidemiological significance beyond their molecular characterization. High-confidence variants such as S450L, H445Y, and H445D confer strong RIF resistance and are linked with poor treatment outcomes, requiring MDR-TB regimens. In contrast, disputed mutations (e.g., L430P, D435Y, H445L, L452P, and I491F) often produce MICs near or below the WHO critical concentration (≤1.0 µg/mL), leading to potential false-susceptible results in phenotypic assays. Despite these borderlines, treatment failure and relapse have been documented in patients harbouring such strains.

The WHO currently recommends that any molecular result indicating RIF resistance should be considered clinically actionable, and patients should be treated with a RR/MDR-TB regimen unless resistance can be confidently ruled out. This guidance is particularly relevant for disputed mutations, which often produce MICs near critical concentrations and produce inconsistent phenotypic DST results. Evidence from clinical cohorts suggests that these mutations can adversely impact treatment response. Vadakunnel et al. reported poorer treatment outcomes among patients with disputed mutations, particularly L430P, despite some isolates testing phenotypically susceptible [41]. Similarly, Lin et al. showed that isolates carrying disputed mutations such as L430P, L452P, H445N, and D435Y experienced treatment failure [40]. These findings underscore that disputed mutations can influence regimen selection, as treating these cases with standard, drug susceptible regimens increase the risk of inadequate drug exposure, treatment failure, and amplification of resistance to MDR-TB. Therefore, disputed mutations should prompt confirmatory sequencing, close clinical monitoring, and early consideration of RR/MDR-TB regimens to prevent misclassification and mitigate subsequent resistance evolution.

Beyond mutation-specific determinants, the clinical impact of RIF-resistant TB is also shaped by host-related vulnerabilities. HIV co-infection, although not directly altering the *rpoB* mutational spectrum, markedly heightens the consequences of delayed or missed detection of RIF resistance. Individuals living with HIV often exhibit markedly reduced sputum bacillary loads, which substantially compromises the sensitivity of smear microscopy and molecular assays, thereby increasing the probability of false-negative or indeterminate diagnostic results [98]. In HIV patients, the burden of drug-resistant TB is substantial. Resistance to rifampicin and quinolones is high and has continued to increase annually, with age and prior TB treatment identified as major contributing factors [99]. Clinically, HIV co-infection further worsens outcomes. Co-infected patients show markedly poorer functional recovery, minimal improvement in physical performance during hospitalization, and significantly higher mortality compared with HIV-negative TB patients [100]. These combined diagnostic and prognostic challenges amplify the clinical implications of *rpoB* variants in high HIV-burden settings and reinforce the need for rapid molecular testing, integrated HIV–TB management, and strengthened resistance-surveillance systems.

Such vulnerabilities highlight the importance of viewing RIF resistance within the wider spectrum of drug-resistant TB, including pathways leading to MDR and XDR. Multidrug-resistant (MDR) and extensively drug-resistant (XDR) tuberculosis further complicate the clinical and epidemiological impact of *rpoB* mutations. The accumulation of resistance-conferring mutations across multiple drug targets reflects the interplay between bacterial genetic adaptability, host immune pressures, and treatment-related factors that shape resistance evolution [101]. RIF resistance often serves as a gateway to broader resistance patterns, as impaired bacterial clearance under suboptimal immune or therapeutic conditions facilitates additional mutations in genes such as *katG*, *inhA*, *gyrA*, or *rrs*. Emerging evidence also suggests that transient drug tolerance manifested through mechanisms such as cell wall thickening or metabolic adaptation, may precede and promote the acquisition of these fixed resistance mutations, accelerating progression toward MDR and XDR phenotypes [102]. These combined mechanisms contribute to poorer treatment outcomes, increased transmission potential, and greater heterogeneity in resistance phenotypes, underscoring the need to interpret rpoB mutations within the wider context of MDR and XDR development.

The global distribution of *rpoB* mutations is shaped by lineage diversity and antibiotic selection pressure, with S450L remaining the most dominant (≈66% of RIF-resistant isolates) [68], particularly among lineage 2 [103] and lineage 4 strains [50]. This pattern illustrates that mutation prevalence is not uniform across Mtb populations but is strongly influenced by lineage structure, with certain lineages, especially Lineage 2 showing a higher propensity to acquire and transmit *rpoB* mutations associated with RIF resistance. Consistent with these global trends, He et al. reported that lineage 2 strains exhibited a higher overall resistance rate (11.87%) compared with lineage 3 and 4 strains (4.45%), although no statistically significant difference was observed in RIF resistance, which occurred in 6.85% of Lineage 2 strains compared with 3.41% of lineage 3 and 4 strains [104].

Recent lineage level analyses further demonstrate that drug resistance phenotypes are influenced by the genetic background of Mtb. In the study by Li et al. marked differences in MIC distributions were observed for INH and AMK between Lineage 2.2 and 2.3 sublineages, whereas no significant differences were detected in RIF MICs [105]. This indicates that the magnitude of RIF resistance is primarily driven by the specific *rpoB* mutation present rather than by lineage background, in contrast to other first and second line drugs whose MIC profiles showed stronger lineage dependent variation.

In addition to these clinical and epidemiological patterns, factors related to treatment implementation further influence the selection and propagation of *rpoB* mutations. Poor medication adherence has been consistently identified as a major driver of drug-resistant tuberculosis [106]. Among patients who develop RIF-resistant TB, resistance is predominantly mediated by mutations within the *rpoB* gene, which determine both the level of resistance and treatment outcomes [41]. Incomplete or irregular intake of first-line anti-TB drugs reduces drug exposure to subtherapeutic levels [107]. This subtherapeutic exposure creates a selective environment in which resistant *rpoB* mutant bacilli are preferentially sustained and propagated, ultimately accelerating the emergence and fixation of RIF resistance within affected populations.

Taken together, these findings suggest that while lineage strongly shapes which *rpoB* mutations emerge and circulate, it does not substantially modify the phenotypic expression of RIF resistance once a mutation is present. This distinction underscores the importance of integrating both mutation level and lineage level information to accurately interpret regional resistance patterns. Variation in reported *rpoB* mutation frequencies across studies may also reflect differences in sampling strategies, lineage composition, diagnostic platforms, and phenotypic DST protocols, all of which introduce methodological heterogeneity. Accounting for these factors within genomic surveillance systems and ensuring regular updates of molecular diagnostic assays will be essential for improving the accuracy of RIF resistance detection and strengthening the clinical management of drug-resistant tuberculosis.

## 7. Conclusions and Future Perspectives

A deeper understanding of *rpoB* mutations from structural mechanisms to diagnostic and clinical implications has refined our view of RIF resistance in Mtb. Although most resistance-conferring mutations occur within the RRDR, the increasing recognition of low-frequency and disputed variants, including those outside this region, underscores the limitations of hotspot-based assays. Lowering the MIC breakpoint and extending phenotypic incubation time can improve detection of borderline resistance, while integrating sequencing-based approaches such as tNGS and WGS will ensure comprehensive identification of clinically relevant mutations. Strengthening these molecular and phenotypic strategies is essential to close current diagnostic gaps and support global control of RIF-resistant and multidrug-resistant tuberculosis.

## Figures and Tables

**Figure 1 pathogens-15-00016-f001:**
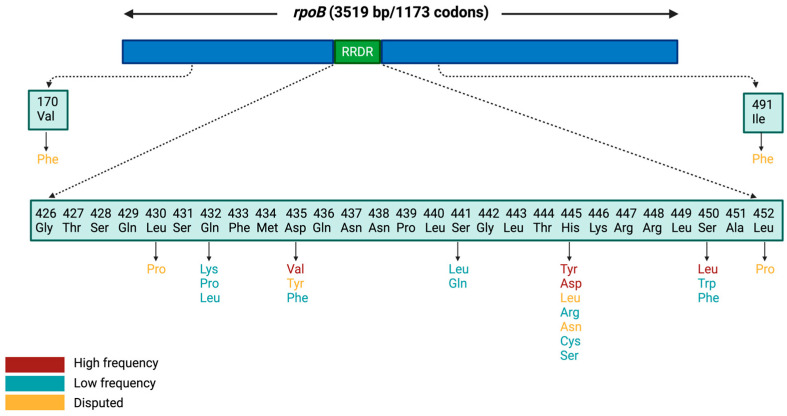
Schematic representation of mutations in the *rpoB* gene. The diagram illustrates the full-length *rpoB* gene, highlighting codons within the RRDR (426–452) as well as codons 170 and 491 located outside the RRDR. The bottom panel shows the RRDR codons with their wild-type amino acids and reported resistance-associated substitutions. Cell colours and arrows indicate mutation categories (high frequency, low frequency, or disputed). RRDR: Rifampicin Resistance-Determining Region.

**Figure 2 pathogens-15-00016-f002:**
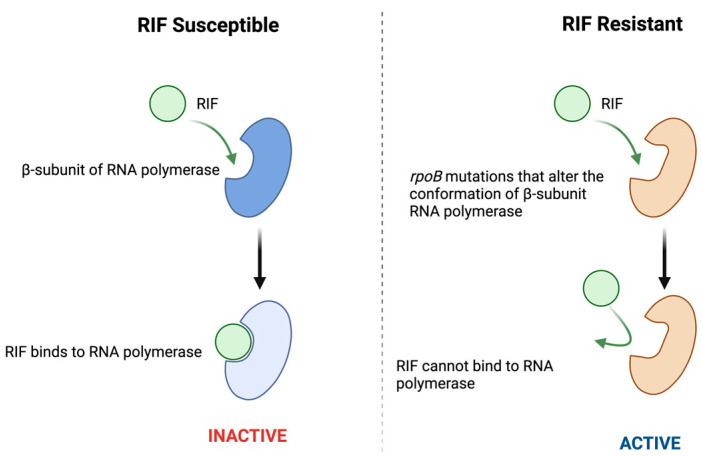
Schematic illustration of rifampicin (RIF) binding to the β-subunit of RNA polymerase in susceptible (**left**) and resistant (**right**) Mtb. Mutations in *rpoB* alter the structure of the RIF-binding pocket, preventing drug binding and preserving RNA polymerase activity. Dark blue represents the unaltered β subunit of RNA polymerase, while light blue represents the RIF-bound RNA polymerase complex, which is inactive and inhibits RNA chain elongation. Orange indicates a conformationally altered β subunit that prevents RIF binding, allowing RNA polymerase activity and bacterial growth.

**Figure 3 pathogens-15-00016-f003:**
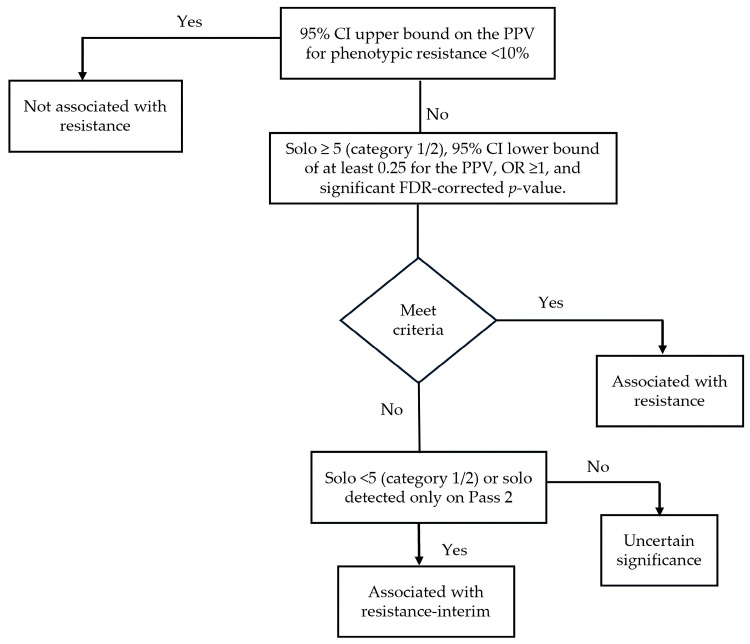
WHO workflow for confidence grading of *rpoB* mutations. Mutations are classified as associated with resistance, associated with resistance–interim, uncertain significance, or not associated with resistance based on PPV confidence intervals, solo occurrence thresholds, and strict WHO statistical criteria. Category 1: WHO-endorsed DST methods using current critical concentrations (L-J, 7H10, 7H11, MGIT). Category 2: Methods using outdated WHO critical concentrations or studies citing WHO guidance without specifying the concentration used.

**Figure 4 pathogens-15-00016-f004:**
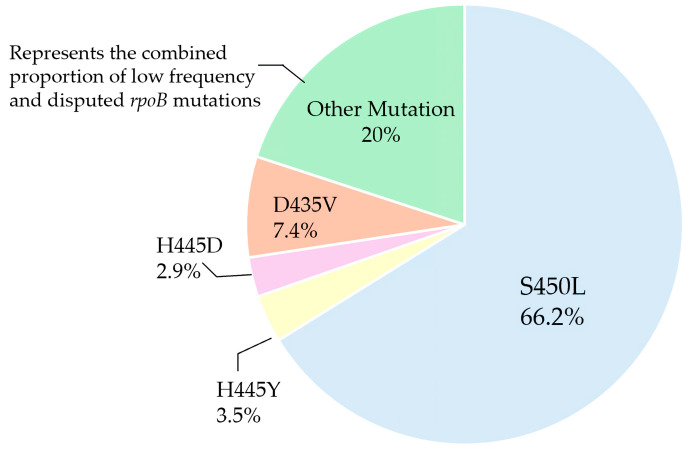
Distribution of *rpoB* mutations associated with RIF resistance, with S450L as the predominant variant. “Other Mutation” denotes the combined proportion of low frequency and disputed mutations.

**Figure 5 pathogens-15-00016-f005:**
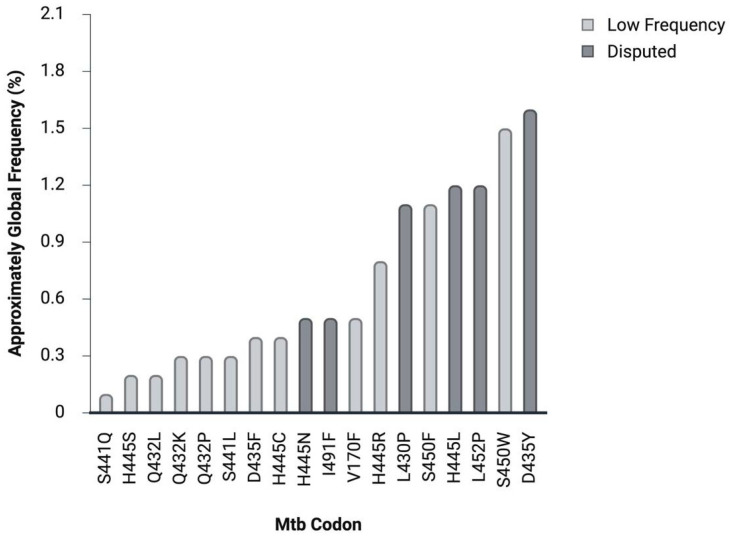
Distribution of selected low frequency and disputed *rpoB* mutations associated with RIF resistance. The chart presents selected low frequency *rpoB* mutations associated with RIF resistance.

**Figure 6 pathogens-15-00016-f006:**
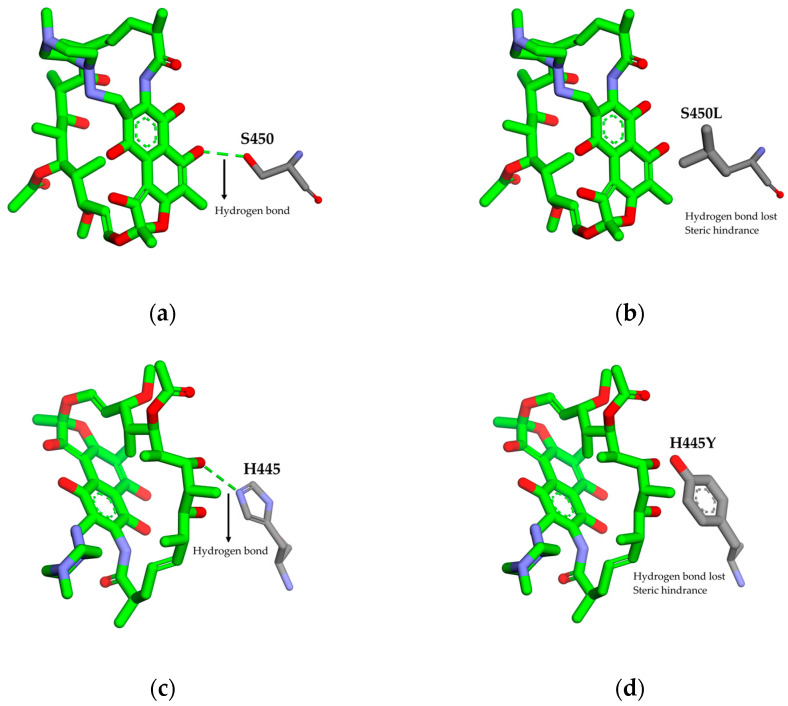
Effects of S450L and H445Y mutations on RIF binding. Panels (**a**–**d**) show that the wild-type residues S450 and H445 form key hydrogen bonds with rifampicin, while the S450L and H445Y substitutions disrupt these interactions and introduce steric hindrance, weakening drug binding.

**Table 1 pathogens-15-00016-t001:** Summary of *rpoB* Mutations in RIF-Resistant Mtb According to WHO Confidence Categories (2021), Global Frequencies, and Levels of Phenotypic Resistance.

Mtb Codon	*E. coli* Codon	WHO Confidence Category (2021)	Approx. Global Frequency	Classification of *rpoB* Mutations	Level of RIF Resistance
S450L	S531L	Associated with resistance	66.2%	High Frequency	High
H445Y	H526Y	Associated with resistance	3.5%	High Frequency	High
H445D	H526D	Associated with resistance	2.9%	High Frequency	High
D435V	D516V	Associated with resistance	7.4%	High Frequency	Moderate-High
D435Y	D516Y	Associated with resistance	1.6%	Disputed	Low-Moderate
S450W	S531W	Associated with resistance	1.5%	Low Frequency	High
L452P	L533P	Associated with resistance	1.2%	Disputed	Low
H445L	H526L	Associated with resistance	1.2%	Disputed	Low
S450F	S531F	Associated with resistance	1.1%	Low Frequency	High
L430P	L511P	Uncertain significance	1.1%	Disputed	Low
H445R	H526R	Associated with resistance	0.8%	Low Frequency	High
V170F	V498A	Associated with resistance	0.7%	Low Frequency	Low
I491F	I572F	Associated with resistance	0.5%	Disputed	Low
H445N	H526N	Uncertain significance	0.5%	Disputed	Low
D435F	D516F	Associated with resistance	0.4%	Low Frequency	Moderate-High
H445C	H526C	Associated with resistance	0.4%	Low Frequency	High
Q432K	Q513K	Associated with resistance	0.3%	Low Frequency	High
Q432P	Q513P	Associated with resistance	0.3%	Low Frequency	High
S441L	S522L	Associated with resistance	0.3%	Low Frequency	Moderate
Q432L	Q513L	Associated with resistance	0.2%	Low Frequency	High
H445S	H526S	Associated with resistance	0.2%	Low Frequency	Low
S441Q	S522Q	Associated with resistance	0.1%	Low Frequency	Low

Data were compiled from the WHO mutation catalogue [68,69]. The level of RIF resistance was inferred from minimum inhibitory concentration (MIC) data and structural modelling studies [31,32,80]. High frequency mutations were defined as those with global prevalence ≥ 2.9%, while low frequency mutations were defined as variants occurring at <2%. These thresholds reflect the distribution reported in the WHO Mutation Catalogue (2021) and do not indicate resistance strength. Several low frequency mutations remain classified as “associated with resistance” by WHO. Disputed mutations: Variants with inconsistent phenotypic resistance, where MIC values often fall near the rifampicin critical concentration despite confirmed resistance-associated changes in *rpoB*.

**Table 2 pathogens-15-00016-t002:** Comparative overview of diagnostic methodologies used for detecting *rpoB* mutations and RIF resistance, including their strengths, limitations, and key analytical characteristics.

Method	Target Region/Genes	Strengths	Limitation	Analytical Parameter	Refs.
MGIT 960	Whole cell response at critical concentration. Growth in liquid media, Critical concentration for RIF resistance = 1.0 µg/mL	Gold standard, WHO-endorsed DST methods	Borderline MICs (≤1.0 µg/mL) may appear susceptible	Time to result: 4–13 day	[85,87]
Löwenstein–Jensen	Whole cell response at critical concentration. Growth in solid media Critical concentration for RIF resistance = 40 µg/mL	Gold standard, WHO-endorsed DST methods, low cost	Slowest method, borderline resistance often missed	Time to result: 4–6 weeks	[85,87]
Xpert MTB/RIF	*rpoB* RRDR (81 bp)	Rapid and strong sensitivity for common RRDR mutations	Only RRDR coverage. Does not detect non-RRDR mutations	Time to result: ~2 h Sensitivity for RIF: 95.3–97.4%	[35,88,89,90]
Xpert MTB/RIF Ultra	*rpoB* RRDR (81 bp)	Improved sensitivity through multi-copy targets (IS6110/IS1081) and lower LOD	Only RRDR coverage. Does not detect non-RRDR mutations	Time to result: ~1.5–2 h Sensitivity for RIF: 94.9–97%	[76,88,90]
Line Probe Assay (LPA)	*rpoB* RRDR (81 bp), katG, inhA	Detects RIF + INH resistance, mutation specific binding patterns	Only RRDR coverage. Does not detect non-RRDR mutations	Time to result: ~6–8 h Sensitivity for RIF: 92.3–95.8% (GenoType MTBDR plus v2)	[77,90,91]
tNGS	Full length *rpoB* or extended regions	Detects RRDR + non-RRDR + disputed variants, high resolution, detects all mutation types	Requires sequencing facilities, cost higher than probe assays	Time to result: 3 days, Sensitivity for RIF ONT: 95.7–97.4% Genoscreen: 97.3–99.0%	[90,91,92]
WGS	Entire genome (including full length *rpoB*)	Most comprehensive, lineage + resistance prediction, identifies emerging variants, full resistance and phylogeny	Highest cost requires advanced bioinformatics. Slower than tNGS	Time to result: 3–4 weeks Sensitivity for RIF: 87.5–100%	[93,94,95]

## Data Availability

The data supporting this review were derived from previously published studies and publicly available sources cited in the manuscript. No new data were generated in this work.

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
