# Peer review of "Beyond Hotspot Mutations: Diagnostic Relevance of High Frequency, Low Frequency, and Disputed rpoB Variants in Rifampicin-Resistant Mycobacterium tuberculosis"

_pathogens, 2025, doi:10.3390/pathogens15010016_

Round 1

Reviewer 1 Report

Comments and Suggestions for Authors

The manuscript is well written, and provides a well-referenced overview of the rpoB mutations associated with rifampicin resistance, highlighting diagnostic implications of high-frequency, low-frequency and disputed mutations. I have the following suggestions for the authors to consider:

1) The methods section needs strengthening, the data sources and compilation approach section is too weak as it does not specify search strategy (databasesm years, search terms, language restrictions etc), inclusion/exclusion criteria, number of articles screened vs included? Is this a narrative, scoping or systematic review?

2) The review is too descriptive lacking some critical analysis for example around the conflicting results across studies (why may that be, what are the implications), how does lineage diversity affect mutation prevalence, what are the strengths and weaknesses of diagnostic tools (perhaps add some diagnostic recommendations), what are the limitations of the WHO2021 mutation catalogue? what is the authors' assessment of clinical outcome inconsistencies linked to disputed mutations?

3) The authors could additionally discuss briefly about the gaps in structural modelling research, and what they would recommend for future research priorities.

4) I would recommend adding a figure showing the RIF-binding pocket

5) In the clinical section the authors could briefly touch on WHO recommendations for RIF-resistant TB with borderline mutations, evidence linking mutation type to treatment outcomes, how disputed mutations affect regimen choice and the risk of amplification to MDR TB.

Author Response

Comments 1: The methods section needs strengthening, the data sources and compilation approach section is too weak as it does not specify search strategy (databases, years, search terms, language restrictions etc), inclusion/exclusion criteria, number of articles screened vs included? Is this a narrative, scoping or systematic review?

Response 1: Thank you very much for this helpful comment. This manuscript is a narrative review; therefore, it does not follow systematic or scoping review methodology. We have revised the Data Sources and Compilation Approach section to clarify this and to describe the key databases, search topics, and literature sources used to guide the selection of relevant studies. These revisions have been added in lines 110–114 of the revised manuscript.

Comments 2: The review is too descriptive lacking some critical analysis for example around the conflicting results across studies (why may that be, what are the implications), how does lineage diversity affect mutation prevalence, what are the strengths and weaknesses of diagnostic tools (perhaps add some diagnostic recommendations), what are the limitations of the WHO2021 mutation catalogue? what is the authors' assessment of clinical outcome inconsistencies linked to disputed mutations?

Response 2: We thank the reviewer for the valuable suggestions. In response, we have strengthened the analytical components of the review by:

(i) discussing potential sources of conflicting findings across studies (lines 508–513),
(ii) addressing lineage-dependent differences in rpoB mutation prevalence (lines 477–486),
(iii) expanding the diagnostic section to include strengths, weaknesses, and practical recommendations (lines 375–384),
(iv) outlining key limitations of the WHO 2021 mutation catalogue (lines 170–173), and
(v) adding our assessment of clinical inconsistencies associated with disputed mutations (lines 423–429 and 435–439).

These revisions have been incorporated into Sections 5 and 6 of the manuscript.

Comments 3: The authors could additionally discuss briefly about the gaps in structural modelling research, and what they would recommend for future research priorities.

Response 3:Thank you for this helpful suggestion. We have now added a brief discussion in the revised manuscript outlining the current limitations in structural modelling of rpoB–rifampicin interactions. These revisions have been added in lines 332–341 and 347-361 of the revised manuscript.

Comments 4: I would recommend adding a figure showing the RIF-binding pocket.

Response 4: Thank you for the suggestion. The figure illustrating the rifampicin-binding pocket has now been added in Section 3 The rpoB Gene and the Rifampicin Resistance Mechanism to improve the clarity of the structural explanation. These revisions have been added in lines 130-133 of the revised manuscript.

Comments 5: In the clinical section the authors could briefly touch on WHO recommendations for RIF-resistant TB with borderline mutations, evidence linking mutation type to treatment outcomes, how disputed mutations affect regimen choice and the risk of amplification to MDR TB.

Response 5:Thank you for the suggestion. We have added a brief explanation in Section 6 addressing WHO guidance on borderline/disputed rpoB mutations and summarizing evidence on their impact on treatment outcomes. These revisions have been added in lines 430-444 of the revised manuscript.

Reviewer 2 Report

Comments and Suggestions for Authors

In this manuscript, the authors examine the diagnostic significance of high-frequency, low-frequency, and disputed rpoB variants associated with rifampicin-resistant Mycobacterium tuberculosis, extending the discussion beyond traditional hotspot mutations. The manuscript is well written and following are some comments.

  1. The authors should contextualize their analysis in relation to the concurrent burden of HIV/AIDS.
  2. They may also consider incorporating relevant information on treatment adherence and its influence on drug resistance in connection with the observed mutations.
  3. Additionally, the discussion should address MDR, XDR, and other resistance patterns, highlighting the interactions among resistance mechanisms, host immunity, and genetic variability.

Author Response

Comments 1:  The authors should contextualize their analysis in relation to the concurrent burden of HIV/AIDS.

Response 1: Thank you for your suggestion We have revised the discussion to contextualize our findings within the burden of HIV/AIDS, highlighting how HIV co-infection affects TB diagnosis, rifampicin resistance detection, and clinical outcomes. These revisions have been added in lines 445–460 of the revised manuscript.

Comments 2: They may also consider incorporating relevant information on treatment adherence and its influence on drug resistance in connection with the observed mutations.

Response 2:Thank you for the suggestion. We have added a brief explanation in the discussion on how poor treatment adherence contributes to the emergence and selection of drug-resistant strains, including rpoB mutations associated with rifampicin resistance, to better contextualize our findings. These revisions have been added in lines 494–503 of the revised manuscript.

Comments 3: Additionally, the discussion should address MDR, XDR, and other resistance patterns, highlighting the interactions among resistance mechanisms, host immunity, and genetic variability.

Response 3: Thank you for the suggestion. We have added a brief discussion on MDR, XDR, and related resistance patterns, emphasizing their interaction with resistance mechanisms, host immunity, and genetic variability. These revisions have been added in lines 461–476 of the revised manuscript

4. Response to Comments on the Quality of English Language

Point 1: The English is fine and does not require any improvement.

Response 1: We thank the reviewer for the positive assessment of the English quality. No changes were required in this aspect.

Reviewer 3 Report

Comments and Suggestions for Authors

  • What is the mechanism by which RIF crosses the Mtb cell wall? A genetic mutation mediated by the pe11/lipX132 genes: How would this mechanism affect the following genes: Rv1169c (pe11/lipX 132), Rv3083 (mymA), Rv3082c (virS), Rv3903c (cpnT)?
  • The following statement is unclear: “Several efflux systems, including Rv0783 and Rv2936, have been shown to contribute to low-level RIF resistance”
  • The introduction would be much easier to follow if it included a figure showing the different mutations of Mtb that lead to RIF resistance.
  • The section: 'Classification of rpoB mutations' would benefit from the addition of a schematic figure illustrating the WHO's proposed classification of these mutations.
  • In addition, the authors could give a brief overview of each one, to make them easier to understand for the reader and to establish a link to the next section.
  • The following paragraph is a repetition of information that was already presented in another section: “The majority of RIF-resistant Mtb isolates contain mutations in the 81-bp RRDR, 167 which includes codons 426–452 (11,13,37). This hotspot corresponds to the RIF-binding pocket of the RNA polymerase β-subunit, where the drug normally binds to inhibit transcription. Alterations in this region induce conformational changes that disrupt the hydrogen bonding and steric interactions required for rifampicin binding (31), while largely preserving normal polymerase function. Such structural disturbances markedly increase the minimum inhibitory concentrations (MICs) for RIF, resulting in high-level resistance (60,66)” The same applies to the introduction section, which makes this review unnecessarily longer.
  • A diagram could be created by the authors to summaries the information in the "High-Frequency Mutations" section, which would make it clearer. This section should provide an overview of the frequency of each mutation rather than just a description. The authors should also explain the various hypotheses that can be used to explain these mutations, along with their respective probabilities of occurrence.
  • In each subsection of the general section: 'Classification of rpoB mutations', a figure could be added outlining the specific mutations to which each subsection refers. This figure could also show the frequency of occurrence and probability of each mutation.
  • It is unclear what the authors mean by: "they may not always be covered by…! In line 229.
  • The authors are encouraged to add a section that briefly discusses the different diagnostic methodologies and their respective advantages and disadvantages. This section should also include a comparative table of the methodologies and their analytical parameters.
  • The limits of low, moderate, or high frequency should be defined by Table 1.
  • To improve clarity, the authors are encouraged to include a figure showing the region of RNA that interacts with the RIF, indicating possible mutations and the predicted outcome of molecular modelling studies.
  • The bibliography is somewhat outdated, and there is a need to update it.

Author Response

Comments 1: What is the mechanism by which RIF crosses the Mtb cell wall? A genetic mutation mediated by the pe11/lipX132 genes: How would this mechanism affect the following genes: Rv1169c (pe11/lipX 132), Rv3083 (mymA), Rv3082c (virS), Rv3903c (cpnT)?

Response 1: We thank the reviewer for this insightful point. We have now expanded the explanation of how cell-wall–associated genes (Rv1169c/pe11, Rv3083/mymA, Rv3082c/virS, and Rv3903c/cpnT) influence rifampicin permeability. A brief mechanistic description is added in the revised introduction with citations. These revisions have been added in lines 136-145 of the revised manuscript.

Comments 2: The following statement is unclear: “Several efflux systems, including Rv0783 and Rv2936, have been shown to contribute to low-level RIF resistance”

Response 2:We thank the reviewer for this insightful point. We clarified the mechanism by which Rv0783 and Rv2936 contribute to low-level rifampicin resistance. These revisions have been added in lines 149–151 of the revised manuscript.

Comments 3: The introduction would be much easier to follow if it included a figure showing the different mutations of Mtb that lead to RIF resistance.

Response 3:Thank you for the suggestion. We have added a new figure in the Introduction summarizing the different mutation of Mtb that lead to RIF resistance. These revisions have been added in lines 66–73 of the revised manuscript

Comments 4: The section: 'Classification of rpoB mutations' would benefit from the addition of a schematic figure illustrating the WHO's proposed classification of these mutations.

Response 4:Thank you for the suggestion We added a schematic figure summarizing the WHO mutation classification for improved clarity in this section. These revisions have been added in lines 184-190 of the revised manuscript.

Comments 5: In addition, the authors could give a brief overview of each one, to make them easier to understand for the reader and to establish a link to the next section.

Response 5: We thank the reviewer for the suggestion. A concise overview of high frequency, low frequency, and disputed rpoB mutations has been added to improve clarity and provide a smoother transition to the subsequent section. These revisions have been added in lines 174-183 of the revised manuscript.

Comments 6: The following paragraph is a repetition of information that was already presented in another section: “The majority of RIF-resistant Mtb isolates contain mutations in the 81-bp RRDR, 167 which includes codons 426–452 (11,13,37). This hotspot corresponds to the RIF-binding pocket of the RNA polymerase β-subunit, where the drug normally binds to inhibit transcription. Alterations in this region induce conformational changes that disrupt the hydrogen bonding and steric interactions required for rifampicin binding (31), while largely preserving normal polymerase function. Such structural disturbances markedly increase the minimum inhibitory concentrations (MICs) for RIF, resulting in high-level resistance (60,66)” The same applies to the introduction section, which makes this review unnecessarily longer.

Response 6: Thank you for the suggestion The redundant RRDR description has been removed, and the section has been streamlined to avoid repetition with the Introduction. These revisions have been added in lines 192-198 of the revised manuscript.

Comments 7:  A diagram could be created by the authors to summaries the information in the "High-Frequency Mutations" section, which would make it clearer. This section should provide an overview of the frequency of each mutation rather than just a description. The authors should also explain the various hypotheses that can be used to explain these mutations, along with their respective probabilities of occurrence.

Response 7: Thank you for the suggestion. We have added a new diagram (Figure 4) to summarize the frequency of high frequency rpoB mutations, providing a clearer overview than text alone. We also included a structural illustration (Figure 6) and a brief explanation of the hypotheses underlying the high occurrence of certain mutations, particularly the S450L and H445Y substitution. These revisions have been added in lines 211-214 and 342-345 of the revised manuscript.

Comments 8:  In each subsection of the general section: 'Classification of rpoB mutations', a figure could be added outlining the specific mutations to which each subsection refers. This figure could also show the frequency of occurrence and probability of each mutation.

Response 8: Thank you for the suggestion .We have added an image listing its frequency. These revisions have been added in lines 259-265 of the revised manuscript.

Comments 9:  It is unclear what the authors mean by: "they may not always be covered by…! In line 229.

Response 9: Thank you for your correction. We rephrased the sentence for clarity and specified which diagnostic assays may not detect particular mutations.

We apologize for the oversight; the sentence in question is located in line 219, not line 229. These revisions have been added in lines 247-250 of the revised manuscript.

Comments 10:  The authors are encouraged to add a section that briefly discusses the different diagnostic methodologies and their respective advantages and disadvantages. This section should also include a comparative table of the methodologies and their analytical parameters.

Response 10: Thank you for the suggestion A new section summarizing the major diagnostic methods, their advantages and limitations, and a comparative table of analytical parameters has been included. These revisions have been added in lines 362-373 of the revised manuscript.

Comments 11: The limits of low, moderate, or high frequency should be defined by Table

Response 11: Thank you for the comment. We have now clarified the limits of the frequency categories directly based on the distribution shown in Table 1.  These revisions have been added in lines 314-317 of the revised manuscript.

Comments 12:  To improve clarity, the authors are encouraged to include a figure showing the region of RNA that interacts with the RIF, indicating possible mutations and the predicted outcome of molecular modelling studies.

Response 12: A figure of the RNA polymerase β-subunit showing the rifampicin-binding region and key mutations has been added. These revisions have been added in lines 129-133 of the revised manuscript.

Comments 13:  The bibliography is somewhat outdated, and there is a need to update it.\

Response 13: Thank you for this observation. We have updated the bibliography by incorporating more recent studies published within the past 3–5 years, including new evidence on rpoB mutation patterns, structural modelling, and diagnostic performance. These additions have strengthened the scientific currency and relevance of the review

Round 2

Reviewer 1 Report

Comments and Suggestions for Authors

The manuscript is now scientifically sound and ready for publication. I have no further comments and I am happy with the authors's revisions.

Reviewer 3 Report

Comments and Suggestions for Authors

The authors' responses to the reviewer's comments were satisfactory.